# Development of DLC-Coated Solid SiAlON/TiN Ceramic End Mills for Nickel Alloy Machining: Problems and Prospects

Sergey N. Grigoriev, Marina A. Volosova *, Sergey V. Fedorov, Anna A. Okunkova, Petr M. Pivkin, Pavel Y. Peretyagin and Artem Ershov

Department of High-Efficiency Processing Technologies, Moscow State University of Technology "STANKIN", Vadkovskiy per. 3A, 127055 Moscow, Russia; s.grigoriev@stankin.ru (S.N.G.); sv.fedorov@stankin.ru (S.V.F.); a.okunkova@stankin.ru (A.A.O.); p.pivkin@stankin.ru (P.M.P.); p.peretyagin@stankin.ru (P.Y.P.); a.ershov@stankin.ru (A.E.)
* Correspondence: m.volosova@stankin.ru; Tel.: +7-916-308-49-00

**Abstract:** The study is devoted to the development and testing of technological principles for the manufacture of solid end mills from ceramics based on a powder composition of α-SiAlON, β-SiAlON, and TiN additives, including spark plasma sintering powder composition, diamond sharpening of sintered ceramic blanks for shaping the cutting part of mills and deposition of anti-friction Si-containing diamond-like carbon (DLC) coatings in the final stage. A rational relationship between the components of the powder composition at spark plasma sintering was established. The influence of optimum temperature, which is the most critical sintering parameter, on ceramic samples' basic physical and mechanical properties was investigated. DLC coatings' role in changing the surface properties of ceramics based on SiAlON, such as microrelief, friction coefficient, et cetera, was studied. A comparative analysis of the efficiency of two tool options, such as developed samples of experimental mills made of SiAlON/TiN and commercial samples ceramic mills based on SiAlON, doped with stabilizing additives containing Yb when processing nickel alloys (NiCr20TiAl alloy was used as an example). DLC coatings' contribution to the quantitative indicators of the durability of ceramic mills and the surface quality of machined products made of nickel alloy is shown.

**Keywords:** diamond-like carbon coating; high-speed milling; nickel alloy; SiAlON; spark plasma sintering; roughness; wear resistance

## 1. Introduction

At present, heat-resistant nickel alloys such as Inconel 718 type are widely used to manufacture critical parts operating under high thermal loads. For example, it can be nozzles and working turbine blades, fairings, et cetera. The high-performance characteristics of parts made of nickel alloys determine the difficulties in their machining, accompanied by increased heat and power loads on the cutting tool [1–3]. The machinability factor for high-temperature nickel alloys is in the order of 0.25 compared to the machining of C45 steel (according to EN 10083-2: 2006). Simultaneously, solid carbide end mills are the most popular and versatile tool for machining aircraft parts. A modern approach to solving the problem of increasing machining nickel alloys' productivity over the past years is the development and use for these purposes of end mills made of tool ceramics [4–6]. Today, large international companies, which are recognized leaders in cutting tools production (Kennametal, Mitsubishi Materials, Iscar, and others), produce solid ceramic end mills on an industrial scale [7–9]. Figure 1a shows an example of industrial use of solid ceramic end mills made of SiAlON material for high-speed milling a turbine blade made of nickel alloy on a multi-axis computer numerical control (CNC) machine.

The main advantage of using ceramic end mills when machining nickel alloys over carbide tools is higher heat resistance. Nickel alloys of the Inconel type begin to soften at cutting temperatures of 800 °C and above (Figure 1b), after which significantly lower

power loads accompany their machining on the tool [1]. These temperatures correspond to cutting speeds above 350 m/min. Tool ceramics can be successfully operated due to their higher heat resistance in this high-speed milling mode. Sintered hard alloys at cutting temperatures over 800 °C lose their hardness rapidly (Figure 1b), which excludes the possibility of their use in high-speed milling nickel alloys using the surface layer plasticization effect [10,11].

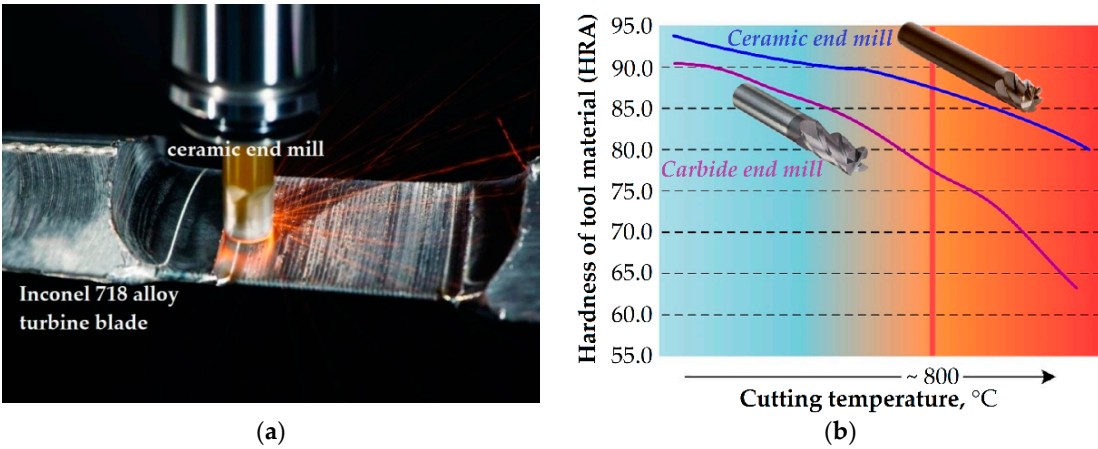

**Figure 1.** An example of industrial use of SiAlON solid ceramic mills in high-speed milling of a nickel alloy turbine blade (**a**) and the variation in surface hardness of end mills produced from ceramics and hard alloy (**b**) as a function of the cutting temperature.

Practice shows that among a wide variety of well-known technical ceramics brands for tool purposes for the solid ceramic end mills manufacturing, the most suitable ceramic material is SiAlON (silicon-aluminum oxynitride), which belongs to the class of ceramics based on silicon nitrides [12–15]. This ceramic consists of three or more phases: α-sialonic, β-sialonic, and amorphous or partially crystallized grain-boundary phases. Sintered ceramics based on α/β-sialons are characterized by a unique combination of even higher hardness than traditional silicon nitride while providing a high level of strength properties. The α-sialon phase has a high hardness that is retained at elevated temperatures, and the β-sialon phase has a high impact strength and fracture toughness. α- and β-sialons are ideally combined, and the ratio between these phases can be quite easily varied when preparing powder compositions (raw materials or precursors) for subsequent sintering [4–6]. It makes it possible to obtain a different set of physical and mechanical properties of the sintered ceramic necessary for the cutting tool's specific operating conditions [16–19].

Moreover, any ceramics, even those obtained using the most advanced technology, are structurally inhomogeneous materials that a priori contain certain defects (such as micropores). The sintering process of ceramic powders involves providing an increased temperature mode and is very technologically complex. In sintering, grain growth can occur; a significant amount of residual glassy phase can be formed, which worsens the hardness and strength of the material during high-temperature operation [20,21]. Excessive porosity can form at a low rate of grain-boundary diffusion and insufficient shrinkage of the powder composition, which sharply worsens the sintered ceramics' fracture toughness [22–24]. It should be borne in mind that a sintered ceramic workpiece, to obtain multi-edge tools, must be subjected to dimensional shaping processing by mechanical methods. One of the main machining methods is grinding with diamond abrasive tools by CNC grinding and sharpening machines. Grinding is also extremely difficult and energy-intensive. Additional difficulties are associated with the reduced electrical and thermal conductivity of ceramics. The physical and technical nature of the diamond grinding process is such that it introduces additional defects (chips, microcracks) into the surface layer of a ceramic workpiece [22,25–27]. Therefore, if a workpiece containing cracks and pores was formed at the sintering stage, they will inevitably become local foci of microfracture

and chips of the cutting part, which will reduce the tool performance. That is why leading researchers and technologists prioritize developments in improving technological processes for sintering ceramics and improving the ceramic powder composition to improve the basic physical and mechanical properties of ceramic blanks with a simultaneous improvement of their machinability.

A promising technological process for sintering SiAlON ceramics is the spark plasma sintering (SPS) method. The sintered powder composition and the used mold are heated by passing high-frequency low-voltage pulses of direct electric current through them. Compared to traditional hot pressing, SPS can significantly reduce the holding time at the maximum temperature (in the order of several minutes). Due to the direct transmission of electric current, it is possible to reduce grain growth rate since it allows high heating and cooling rates of the ceramics [28–30].

Large reserves for improving the physical and mechanical properties of ceramics based on SiAlON are introducing alloying components. For example, the introduction of up to 20 wt.% TiN nanoparticles into a powder composition based on $\alpha/\beta$-sialons can provide an optimal ratio between hardness and impact toughness, improve electrical and thermal conductivity, crack resistance, and machinability [31–36]. Besides, stabilization during unstable high-temperature phases sintering, which is carried out by alloying with certain stabilizing elements based on rare-earth metal oxides, is of great importance. Today, Nd, Sm, Gd, Dy, Y, and Yb elements introduced in the form of nanoparticles into a powder composition with a volume of up to 7 wt.% are sufficiently studied and have proven a certain efficiency in SiAlON production [37–40]. World manufacturers, currently industrializing solid ceramic end mills, sinter powder compositions based on $\alpha/\beta$-sialons with stabilizing additives $Y_2O_3$ and $Yb_2O_3$ (the latter option is recognized as more effective). The introduction of alloying rare-earth metals yttrium and ytterbium complicates powder composition preparation and sintering and increases the cost of an already expensive end product with apparent positive aspects. For example, the average selling price per unit for a one-piece ceramic four-flute endmill with a diameter of 10 mm is over 500 Euro. Such a high cost is partly offset by a manifold increase in productivity and durability during operation. Simultaneously, such a tool, compared to hard alloy, is not suitable for subsequent regrinding after reaching the limit wear by the cutting part. This constrains the broader distribution of solid ceramic end mills in mechanical engineering. Today, such a tool is not publicly available for manufacturing enterprises, and exploratory research aimed at finding approaches to simplifying solid ceramic end mills manufacturing technology and reducing their manufacturing costs is relevant.

One of the approaches to increasing the wear resistance of one-piece ceramic end mills can be various coatings deposition on their working surfaces based on complex nitrides and diamond-like carbon structures. This approach is based on the fact that under various loading conditions of the tool cutting part, its surface layer is the most loaded, and in many respects, this layer determines the wear resistance under the influence of external loads [41–44]. The coatings' effectiveness in increasing the wear resistance of solid ceramic mills in machining nickel alloys has not been experimentally studied until now. Simultaneously, in assessing the effectiveness of this approach to improve assembled turning tools and end mills equipped with ceramic plates, researchers do not have one single point of view [25,26]. Some experts are not inclined to consider coatings' deposition as a viable approach to improve ceramic tools' wear resistance. It can be assumed that such conclusions are the result of overestimated expectations from coatings applied to ceramics. One should not expect an effect comparable to that achieved for carbide tools (multiple increases in durability) from coatings. Initially, it is necessary to proceed from the fact that the role of coatings deposited onto ceramics is specific, and even a twofold increase in the ceramics' resistance should be considered a good result. Many experimental works demonstrate that with rationally chosen deposition technology, architecture, and coatings' composition, their particular possibilities for increasing resistance are quite real and significant [42–46]. For example, the authors of this work have achieved an increase in

resistance during turning and milling hardened steels by 1.4–1.9 times in previous studies when vacuum-plasma coatings' deposition onto cutting inserts made of tool ceramics based on $Al_2O_3$ + TiC and $Al_2O_3$ + SiC [25,26,42], which was a consequence of the effect of coatings on the properties of the surface and the surface layer of ceramic plates. The changes were manifested in the form of a decrease in the friction coefficient between the tool and the processed material, some improvement in strength characteristics, smoothing and reduction of the height of surface microroughness, and a decrease ("healing") of ceramic plates' surface defects that were formed during diamond grinding [25,47]. Thus, there are prerequisites for the fact that the approach to increasing the wear resistance of solid ceramic mills, based on various coatings' deposition onto their working surfaces that can also have a positive effect.

Within the framework of this work, a set of problems was solved, which can be formulated as follows:

1.  development and implementation of laboratory technology for the manufacture of solid ceramic end mills from SiAlON ceramics, including spark plasma sintering powder composition and subsequent diamond grinding of sintered ceramic blanks for shaping the cutting part of end mills of the required geometry;
2.  the establishment of a rational ratio between the components of the powder composition α-SiAlON, β-SiAlON, and TiN during spark plasma sintering ceramic blanks and the choice of the sintering process optimal temperature, as the most critical parameter of the technological process (expensive stabilizing additives used in industry in the production of SiAlON ceramics were deliberately excluded);
3.  using the example of promising Si-containing diamond-like carbon (DLC) coatings to show their effect on the properties of the surface and the surface layer of sintered ceramic samples; to carry out a comparative analysis of the effectiveness of two cutting tool options—developed samples of experimental end mills made of SiAlON/TiN and industrial samples of commercial ceramic mills based on SiAlON, alloyed with stabilizing additives containing Yb; and to establish the contribution of DLC coatings to the quantitative indicators of the resistance of ceramic end mills and the surface quality of workpieces made of nickel alloy (for example, NiCr20TiAl alloy, standard EN 10269).

## 2. Materials and Methods

### 2.1. Solid Ceramic End Mill Design and Geometry

Figure 2a shows a drawing, and Figure 2b is a general view of experimental solid ceramic end mills under development and field testing. Table 1 provides detailed information on the design and geometric parameters of the cutting tool. The experimental samples of end mills produced in this framework of the current study were compared with ceramic tools of a similar design and geometry, which are now commercially manufactured from SiAlON-based ceramics, including Yb-containing stabilizing phases (commercial ceramic mills), and adopted by the authors as a standard.

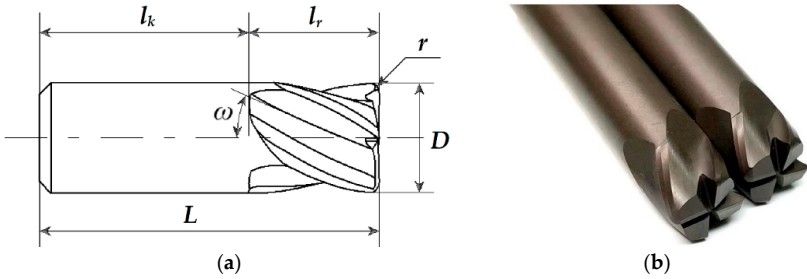

|  (a)  |  (b)  |

**Figure 2.** Drawing (**a**) and general view (**b**) of experimental samples of solid ceramic end mills made by the team of authors.

**Table 1.** The geometric design of solid ceramic end mills for development and testing purposes.

| Tool Parameter | Value |
| --- | --- |
| End mill diameter $D$, mm | 10 |
| Number of teeth | 4 |
| Overall length $L$, mm | 48 |
| Fastener (shank) length $l_k$, mm | 40 |
| Cutting length $l_r$, mm | 8 |
| Vertex radius $r$, mm | 1.2 |
| Flute helix angle $\omega$, degree | 30 |

*2.2. Technological Algorithm of Solid Ceramic End Mill Manufacturing*

The technological algorithm for solid ceramic end mills manufacturing developed by the study's authors is visualized in Figure 3. It includes the following technological operation steps when the finishing operation of coating is not indicated on the diagram:

1. preparation of powder composition based on $\alpha$-SiAlON, $\beta$-SiAlON, and TiN nanoparticles;
2. high-temperature spark plasma sintering of a powder composition in graphite dies, which defines the shape of a future workpiece (a ceramic disk), using the force action of the upper and lower punches and passing powerful DC pulses through the composition to be sintered;
3. cutting a sintered ceramic disk into quadrangular parallelepipeds on a disk abrasive cutting machine (dimensions are set based on the main overall dimensions of ceramic rods and allowances for subsequent grinding);
4. shaping a ceramic rod on a centerless grinding machine;
5. shaping a ceramic rod on a CNC tool sharpening and grinding machine: helical flute, undercut face, and toroidal cutting edge (margin);
6. deposition of vacuum-plasma coatings on solid ceramic end mills.

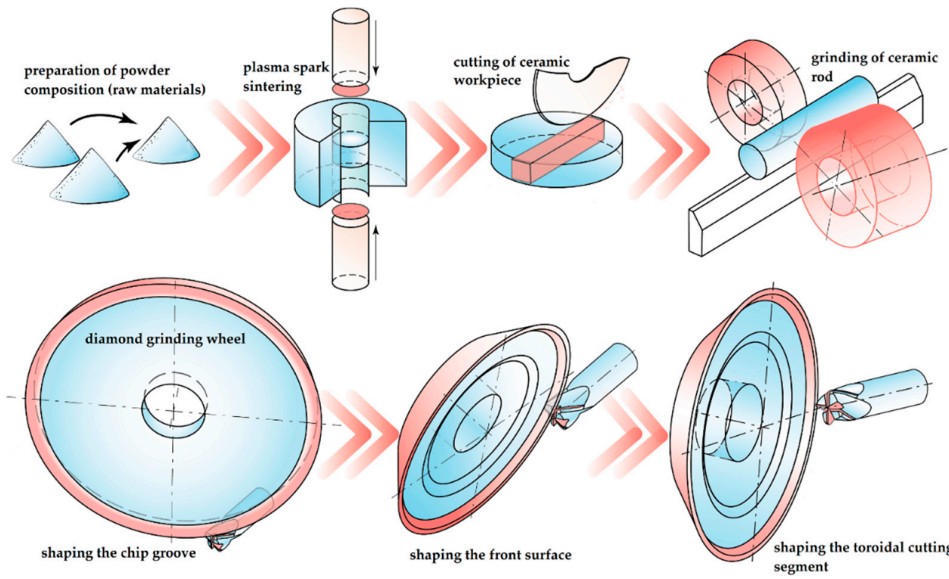

**Figure 3.** Technological algorithm for manufacturing solid ceramic end mills.

*2.3. Powder Compositions' Preparation and Sintering Ceramic Blanks*

Powders produced by Plasmotherm (Moscow, Russia) were used as initial ceramic precursors: $\alpha$-SiAlON and $\beta$-SiAlON with a particle size of $1 \pm 0.5$ μm and TiN additives with a particle size of 15–175 nm. The colloidal method was used for mixing the ceramic powder composition since it allows mixing with minimal time and energy costs.

Three suspension options based on powder compositions were prepared in a ball mill for 24 h using isopropyl alcohol and ceramic grinding bodies in a polyethylene container. The choice of composition content was made taking into account the data of previous

works conducted by authoritative researchers and based on the professional experience of the current study authors [22,31,32]:

1. Option 1: ceramic base α–β SiAlON 80 wt.% (α-SiAlON 90 wt.% + β-SiAlON 10 wt.%) + TiN 20 wt.%; composition code 80% (90α10β) + 20% TiN;
2. Option 2: ceramic base α–β SiAlON 90 wt.% (α-SiAlON 90 wt.% + β-SiAlON 10 wt.%) + TiN 10 wt.%; composition code 90% (90α10β) + 10% TiN;
3. Option 3: ceramic base α–β SiAlON 80 wt.% (α-SiAlON 70 wt.% + β-SiAlON 30 wt.%) + TiN 20 wt.%; composition code 80% (70α30β) + 20% TiN.

The resulting suspensions were dried using a vacuum drying oven, VO400, at +50 °C for 24 h (Memmert GmbH + Co. KG, Schwabach, Germany). After drying, the powders were sieved using a vibration machine (NL 1015X/010, NL Scientific, Klang, Malaysia), sieve laboratory control of 2 ± 0.5 μm, and $Si_3N_4$ ceramic grinding bodies [48].

After completing all the preparatory operations, the powder compositions were subjected to spark plasma sintering on a technological unit manufactured by FCT Systeme GmbH (Effelder-Rauenstein, Germany) (Figure 4).

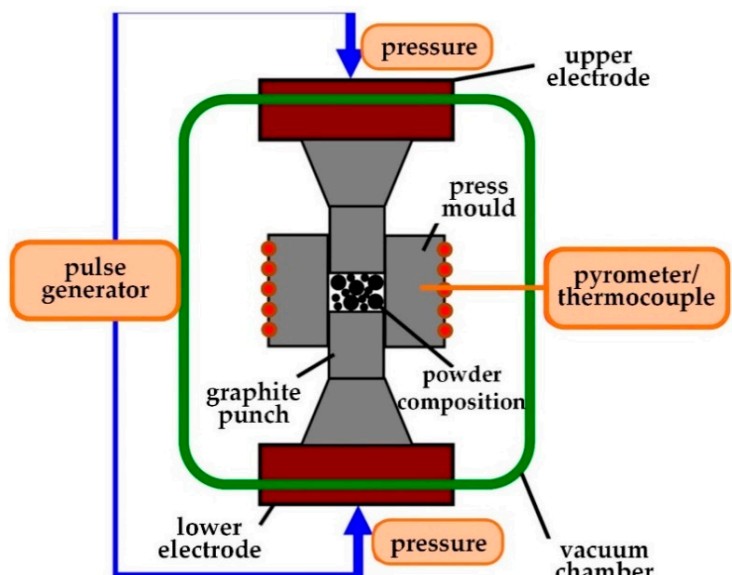

**Figure 4.** A schematic diagram of the ceramic composition's spark plasma sintering.

Ten disk-shaped samples of 3.0 mm thickness and 20.0 mm diameter were made from each version of the powder composition. Further, they were subsequently used to assess sintered samples' main physical and mechanical properties. The ceramic composition that exhibits the best results was subsequently used for coating and tribological tests. The sintering pressure was constant for all types of materials and amounted to 80 MPa, and the holding time at the maximum temperature was 30 min. The sintering temperature was varied from 1600 to 1750 °C with a pitch of 50 °C. These indicators were selected based on the analysis of technical literature [28,30,32] and preliminary experiments carried out by the authors of the current work with spark plasma sintering ceramics based on SiAlON. During sintering, the pressure value was considered rational because its effect on the final samples' mechanical properties was described in a previously published study by the authors of [49]. After determining the powder composition's optimal version and choosing a rational sintering temperature mode, disk-shaped ceramic blanks with 10.2 mm thickness and 80.0 mm diameter were made, from which ceramic rods were obtained.

### 2.4. Diamond Sharpening Solid Ceramic End Mills

Multi-stage diamond sharpening was performed on a Helitronic Micro multi-axis machine from Walter Maschinenbau GmbH (Tübingen, Germany) to manufacture solid

end mills from ceramic rod-blanks with the required design and geometric parameters. The sequence of the main operations of forming ceramic cutters is illustrated in Figure 5.

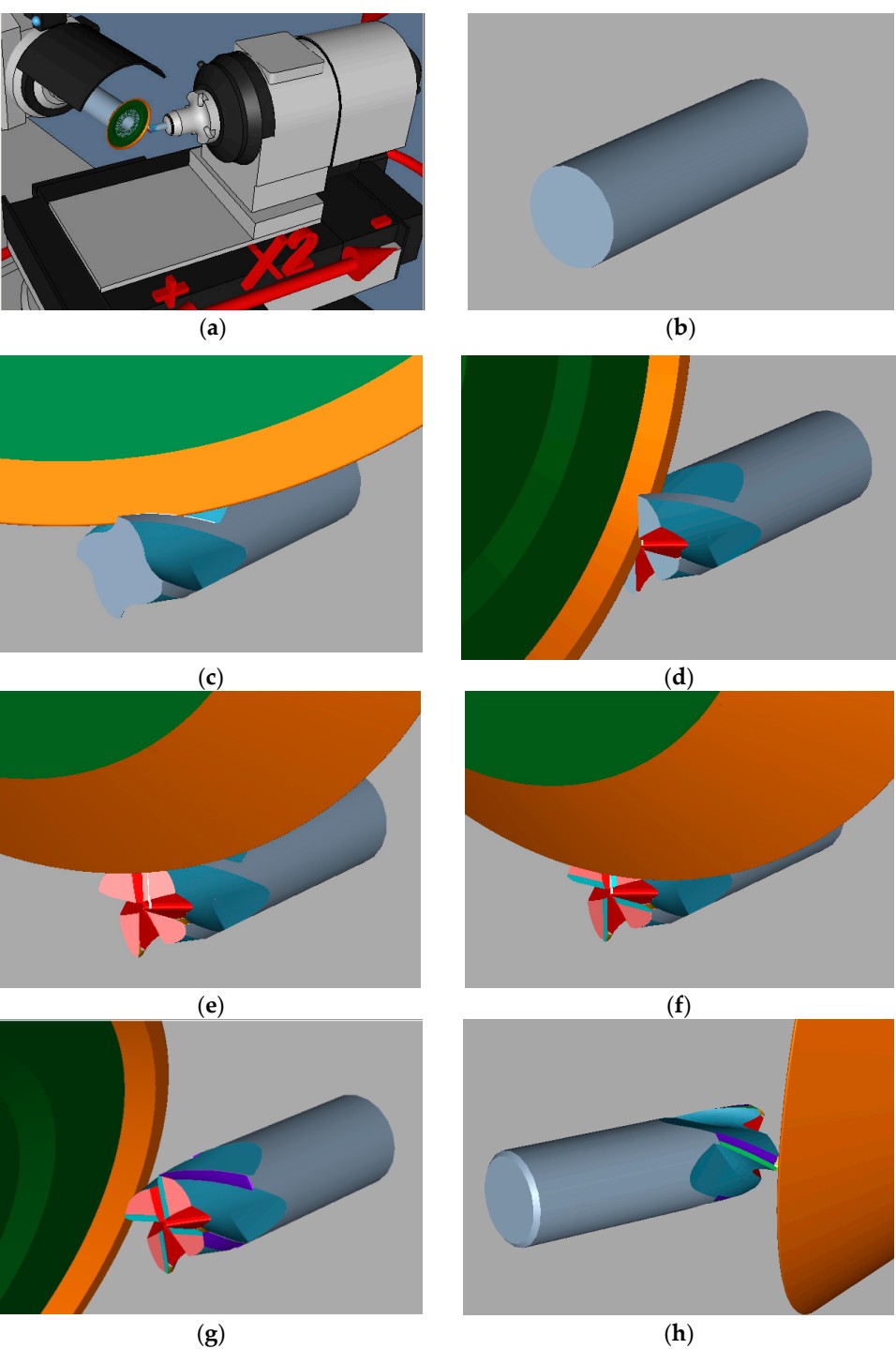

**Figure 5.** The main stages sequence for diamond sharpening solid ceramic end mill: (**a**) the relative position of the machine spindle and the workpiece when forming a ceramic end mill; (**b**) a ceramic rod; (**c**) grinding the helical chip grooves with a tapered disk; (**d**) grinding the rake face at the end with a cup-bevel wheel; (**e**) grinding the secondary relief surface at the end with a cup-bevel wheel; (**f**) grinding the primary relief surface at the end with a cup-bevel wheel; (**g**) grinding the secondary relief surface at the periphery with the cup-bevel wheel; and (**h**) grinding the secondary relief surface on the transitional radius section with the cup-bevel wheel.

The information on chosen cutting modes and the type of grinding wheels used are given in Table 2. The given technological operations and processing modes were selected based on ensuring high-precision execution of ceramic end mills, their profile, geometric dimensions, and high cleanliness of the tool surfaces (0.32–0.5 μm in $R_a$ surface roughness parameter). Particular attention was paid to quality control of the cutting edges of ceramic end mills and minimization of micro-chips that significantly reduce the tool life. The manufactured end mills' control was carried out on a non-contact precision measuring device, Helicheck Plus, manufactured by Walter Maschinenbau GmbH (Tübingen, Germany).

**Table 2.** Modes of ceramic end mill diamond grinding and the type of used grinding wheels.

| Diamond Sharpening Stage | Grinding Wheel Type | Cutting Speed, m·s⁻¹ | Feed, mm·min⁻¹ |
|---|---|---|---|
| Grinding the helical flute | Organic bonded diamond tapered wheel (1V1) | 17 | 10 |
| Grinding the rake face at the end–undercuts at the transitional radius section | | 20 | 7 |
| Grinding the rake at the transitional radius section | Organic bonded diamond cup-tapered wheel (12V9) | 22 | 50 |
| Grinding the secondary relief surface at the end | | 25 | 30 |
| Grinding the primary relief surface at the end | | 25 | 35 |
| Grinding the secondary relief surface on the transitional radius section | | 25 | 30 |
| Grinding the secondary relief surface at the periphery | | 25 | 30 |
| Grinding the primary relief surface on the transitional radius section | | 28 | 40 |
| Grinding the primary relief surface at the periphery | | 25 | 20 |

### 2.5. Vacuum Plasma Coating Composition Selection and Deposition on Solid Ceramic End Mills

The choice of coating deposition method for SiAlON ceramic samples was made in favor of the well-proven technology of physical coating deposition of an evaporated material from a vacuum-arc discharge plasma. This technology makes it possible to form multicomponent coatings of the required composition with high productivity and reproducibility [50–55]. The coating was carried out on a STANKIN-APP technological unit prototype (MSTU Stankin, Moscow, Russia). Figure 6 shows a schematic diagram of a technological unit. The unit provides deposition of various compositions' coatings by varying the composition of the gas mixture supplied to the vacuum chamber through a multi-channel gas injection system and replacing cathodes, the outer surface of which has the shape of a right cone frustum. The formation of plasma by a vacuum arc discharge at a reduced pressure of various gases is accompanied by the appearance of a micro-droplet fraction and a neutral atomic and molecular component of the cathode material's erosion products [56–61]. The unit is equipped with a plasma flow filtration system to eliminate this drawback. A powerful electromagnetic field deflects ions, and microdroplets and neutral particles are captured and removed from the chamber, making it possible to form higher quality coatings on the ceramic samples' surfaces.

Three variations of coatings for deposition on SiAlON ceramic samples, which should contact with nickel alloys, were chosen based on the results of the authors' last works [62–70]: (CrAlSi)N—single-layer coating; (TiAl)N—sandwich-type multilayer coating; and (CrAlSi)N/ DLC—diamond-like carbon two-layer coating (this coating was developed by Platit AG, Selzach, Switzerland). The total thickness of the coatings was in the range of 3.5–3.9 μm. The ceramic samples were thoroughly cleaned in an ultrasonic tank using a soap solution at a temperature of 60 °C for 20 min and in alcohol for 5 min before coating deposition.

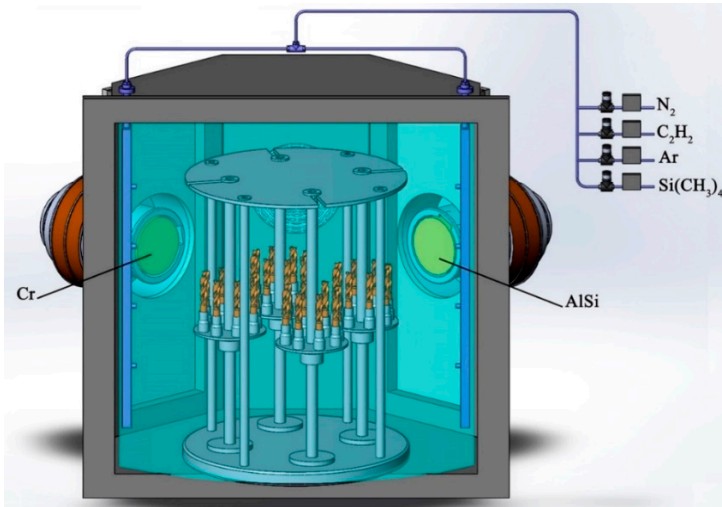

**Figure 6.** A technological unit's principal scheme for coatings' deposition on ceramic samples is using physical deposition of evaporated material from the vacuum arc discharge plasma.

Tribological tests were carried out on disk-shaped sintered specimens with three coating options to assess the prospects of the selected compositions' coatings to increase the wear resistance of SiAlON ceramics. Considering that the coated ceramic tool is expected to operate at elevated cutting temperatures, the tests were carried out both at room temperature and heated to 800 °C. The results of the high-temperature tests were of crucial importance for the selection of the coating, which was subsequently applied to the samples of solid ceramic end mills. The tests were carried out on a TNT tribometer by Anton Paar TriTec (Corcelles-Cormondrèche, Switzerland) using the "ball-on-disk" method with the rotation of the sample relative to a stationary counter body (nickel alloy ball), set at a distance relative to the rotation axis of the sample [71]. Table 3 shows the results of tribological tests with a friction length of 250 m, an applied load of 1 N, and a sliding speed of 10 cm/s.

**Table 3.** Results of preliminary tribological tests of 80% (90α10β) + 20% TiN ceramic specimens with different coatings.

| Specimen | Total Coating Thickness, μm | Testing Temperature, °C | Average Coefficient of Friction (μ) | Wear Track Depth, μm | Ball Wear Diameter, μm |
|---|---|---|---|---|---|
| SiAlON without coating | – | 20<br>800 | 0.49<br>0.81 | 1.4<br>2.2 | 520<br>1000 |
| SiAlON with (CrAlSi)N coating | 3.6 ± 0.1 | 20<br>800 | 0.45<br>0.96 | 0.33<br>2.4 | 195<br>1000 |
| SiAlON with (CrAlSi)N/DLC coating | 3.6 ± 0.1 | 20<br>800 | 0.12<br>0.4 | 0.29<br>1.7 | 165<br>860 |
| SiAlON with (TiAl)N multilayer coating | 3.8 ± 0.1 | 20<br>800 | 0.41<br>0.80 | 0.36<br>2.1 | 180<br>950 |

The coating choice was made in favor of a (CrAlSi)N/DLC two-layer coating based on the experimental data obtained (hereinafter referred to as the DLC coating) since it demonstrated the lowest coefficient of friction and wear-track depth in contact with the counter body during high-temperature heating. It makes it possible to expect that this coating will reduce the intensity of the frictional and adhesive interactions of the cutting tool during milling nickel alloys.

The DLC coating deposition process on solid ceramic end mills (experimental and commercial end mills), implemented in the unit shown in Figure 6, was based on several innovative solutions of the current work's authors and well-known solutions of Platit AG

(Selzach, Switzerland) and provided for the sequential implementation of the complex of the steps listed below:

1.  heating of the tool: pressure in the chamber of 0.03 Pa, temperature in the chamber of 500 °C, table rotation speed of 5 rpm, and holding time of 60 min;
2.  cleaning with argon ions: pressure in the chamber of 2.2 Pa, temperature in the chamber of 500 °C, rotation speed of the table of 5 rpm, negative bias voltage on the table of 800 V, arc current at the Cr cathode of 90 A, and holding time of 20 min;
3.  sublayer (CrAlSi)N deposition: a gas mixture of 5% (Ar) and 95% ($N_2$), pressure in the chamber of 0.9 Pa, temperature in the chamber of 500 °C, table rotation speed of 5 rpm, negative bias voltage on the table of 400 V, arc current at the AlSi cathode of 100 A, arc current at the Cr cathode of 100 A, and holding time of 70 min;
4.  functional DLC coating deposition, including:

    (1) formation of a gradient layer: a gas mixture of 20% (Ar), 73% ($N_2$), and 7% ($Si(CH_3)_4$), pressure in the chamber of 1.5 Pa, temperature in the chamber of 180 °C, rotation speed table of 5 rpm, negative bias voltage on the table of 500 V, and holding time of 20 min;

    (2) formation of a diamond-like carbon layer: a gas mixture of 2% ($Si(CH_3)_4$), 55% (Ar), and 43% ($C_2H_2$), pressure in the chamber of 0.8 Pa, process temperature of 180 °C, table rotation speed of 5 rpm, negative bias voltage on the table of 500 V, and holding time of 100 min.

*2.6. Evaluation of Physical, Mechanical and Surface Properties of Ceramic Materials and Coatings*

The hardness of sintered ceramic compositions based on SiAlON was determined at a load of 2 kg by the Vickers pyramid indentation method on a QnessQ10A universal microhardness tester manufactured (Qness GmbH, Mammelzen, Germany).

The sintered samples' density was estimated using the hydrostatic weighing technique based on Archimedes' law comparing the mass of ceramic samples obtained in air and liquid under normal conditions. The measurements were carried out in distilled water on a GR-300 high-precision analytical balance (A&D Company, Tokyo, Japan).

The samples' strength characteristics were measured by 3-point bending tests at room temperature on an AutoGraph AG-X universal testing machine (Shimadzu, Kyoto, Japan). The loading rate during the tests was 0.5 mm/min, and the maximum displacement was 40 mm. Fracture toughness was also assessed using AutoGraph equipment using a similar loading pattern. A one-sided notch perpendicular to its longitudinal axis was made on ceramic samples' surface with a diamond disk. The sample was installed with a notch downwards and loaded.

The morphology and structure of sintered ceramic specimens and DLC coatings were studied using a VEGA 3 LMH scanning electron microscope (SEM) (Tescan, Brno, Czech Republic) equipped with an Oxford Instruments INCA Energy energy-dispersive X-ray spectroscopy (EDX) system. The sample atoms under study were excited and emitted X-rays characteristic of each chemical element with an electron beam's assistance in the EDX process. When studying the energy spectrum of the indicated radiation using a specialized program, data on the samples' qualitative and quantitative elemental composition were obtained. The program identifies an element and even reveals hidden elements in the sample using the point analysis mode. Transmission electron microscopy (TEM) and selected area diffraction (SAED) studies of the surface layer of ceramic samples with DLC coatings were carried out on JEM-2100F equipment manufactured by JEOL (Tokyo, Japan). The samples were prepared according to the standard probe techniques using Opal 410, Jade 700, and Saphir 300 sample equipment (ATM, Haan, The Netherlands). An epoxy resin with quartz sand was used as a filler for SEM samples. The samples were coated with gold by Quorum 150T ES (Laughton, UK). An instrument controlled the thickness of the coating of less than ~30 μm. TEM samples were polished, cut off 100 nm by an ARTOS 3D Ultramicrotome (Leica, Wetzlar, Germany), and sputtered with gold.

An experimental technique and a special stand, described and tested by the authors in [25], were used to assess the effect of coatings on the resistance of sintered ceramic specimens to fracturing under the action of external loads. The stand was a device in the form of a massive parallelepiped with a groove and step. Sintered ceramic specimens in the form of 20.0 mm diameter and 3.0 mm thickness disks were installed on the step with a small gap. During the experiment, a force was applied to the punch, rigidly fixed on the INSTRON universal testing machine, until the plate broke. A mandatory requirement during the test was to ensure strict reproducibility of the test process, particularly the identity for the entire group of samples of the place of application of force from the punch to the ceramic sample. In the experiment's course, the diagram "force – the punch movement along the z-axis" was recorded, and the critical force at which the destruction occurred was recorded.

The surface microroughness of the sintered ceramic samples before and after DLC coatings' deposition was evaluated on a DektakXT stylus profilometer (Bruker, Billerica, MA, USA).

### 2.7. Performance Testing of Solid Ceramic End Mills

A cylindrical workpiece with a diameter of 40 mm from a heat-resistant nickel alloy NiCr20TiAl [72] was used as the processed material to carry out comparative performance tests of experimental ceramic end mills and commercial ceramic mills. The chemical composition of the workpiece is shown in Table 4. The hardness (HRC) of the processed material was 34 units, and the ultimate strength is 1150 MPa.

**Table 4.** Chemical composition (wt.%) of processed nickel alloy NiCr20TiAl.

| Element | Ni | Cr | Ti | Al | Fe | Si | Co | Mn | Cu | C | Rest |
|---------|----|----|----|----|----|----|----|----|----|----|------|
| Wt.% | 73 | 20 | 2.5 | 1.0 | 1.0 | 0.6 | 0.5 | 0.4 | 0.2 | 0.07 | 0.73 |

The tests were carried out on a 5-axis turning-milling machining center CTX beta 1250TC (DMG, Hüfingen, Germany). The tool and part clamping system's high rigidity minimizes the risks of accidental brittle fracture of the cutting part of ceramic end mills during resistance tests. A SCHUNK GZB-S Ø20/Ø10 adapter sleeve and an SDF-EC bt40 Ø20 HSK-A63 hydraulic arbor to fix the end mills were used, which allow the tool to be securely clamped with high positioning accuracy.

The tool's tests were carried out according to the program written in the CAD/CAM system GeMMa-3D (LLC STC GeMMa, Zhukovsky, Moscow, Russia) when implementing the strategy of machining the plane of the workpiece end face with an end mill when the cutter moves along a spiral. Figure 7a shows a general view of the ceramic end mill's location relative to the nickel alloy NiCr20TiAl workpiece during testing, and Figure 7b visualizes the processing strategy. The tests were carried out under the following cutting modes: cutting speed $V_c$ = 376.8 m/min (rotation frequency $n$ = 12,000 rpm), feed $S$ = 1500 mm/min, and feed per tooth $S_t$ = 0.031 mm/tooth. The scheme of "dry" processing without cutting fluids was used.

The wear area's critical size $h_f$ along the end mill tooth's flank face equal to 0.4 mm was taken as the criterion for the loss of performance (failure) of ceramic end mills (Figure 8). The tool's wear resistance was determined as the time of milling until the end mill reached critical wear (when this value was exceeded, the cases of spalling and chipping of the cutting part increased many times). Each end mill sample was researched optically every minute during milling. The performance tests continued until $h_f$ reaches the range of 0.4–0.5 mm to demonstrate the dramatic tendency in tool wear. A metallographic optical microscope of the Stereo Discovery V12 model (Carl Zeiss Vision GmbH, Jena, Germany) was used to quantify wear with a measurement accuracy of ±0.025 mm, which is 5.00–6.25% of the measured value $h_f$ and less than standard accuracy tolerance. The wear area was monitored on each of the four teeth of the ceramic end mills that were tested. The arithmetic

mean values were calculated based on the data obtained, which were used to plot the wear curves of ceramic end mills.

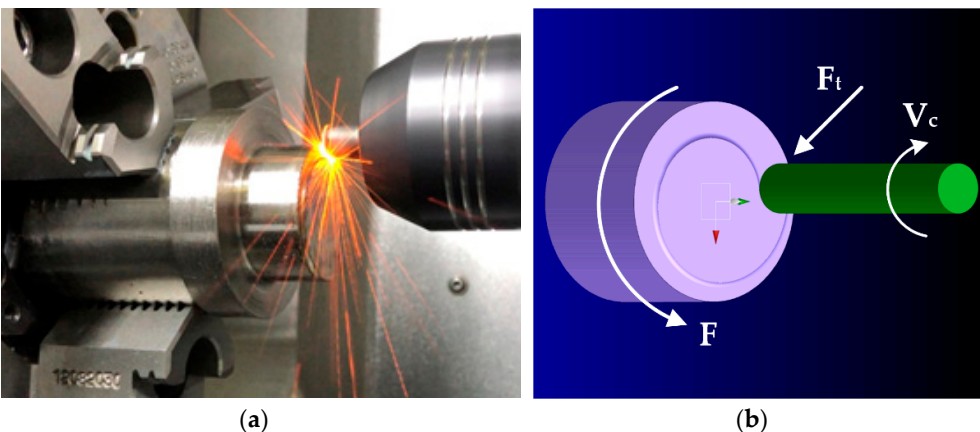

| (**a**) | (**b**) |

**Figure 7.** Strategy for resistance testing of solid ceramic end mills: (**a**) general view of the ceramic end mill location relative to the NiCr20TiAl nickel alloy workpiece during testing; and (**b**) processing strategy visualization.

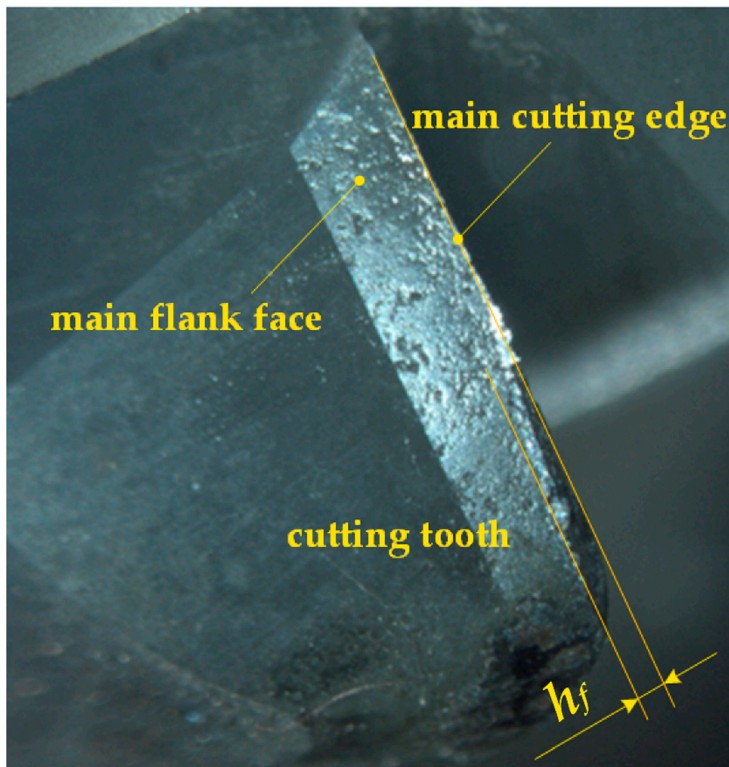

**Figure 8.** The wear zone location on the flank face of the ceramic end mill tooth and the $h_f$ area (land) to measure the quantified wear.

A Surftest SJ-410 portable profilometer (Mitutoyo, Kawasaki, Japan) was used for a quantitative and qualitative assessment of a nickel alloy workpiece's surface roughness after machining with various ceramic end mills.

## 3. Results and Discussion

### 3.1. Structure and Properties of Sintered Ceramic Blanks

Figure 9 shows the experimentally obtained data on the effect of various powder compositions (1) 80% (90α10β) + 20% TiN, (2) 90% (90α10β) + 10% TiN and (3) 80%

(70α30β) + 20% TiN and spark plasma temperatures in the range of 1600–1750 °C on the basic physical and mechanical properties of sintered ceramic samples: percentage theoretical density (Figure 9a), hardness (Figure 9b), crack resistance (Figure 9c), and flexural strength (Figure 9d).

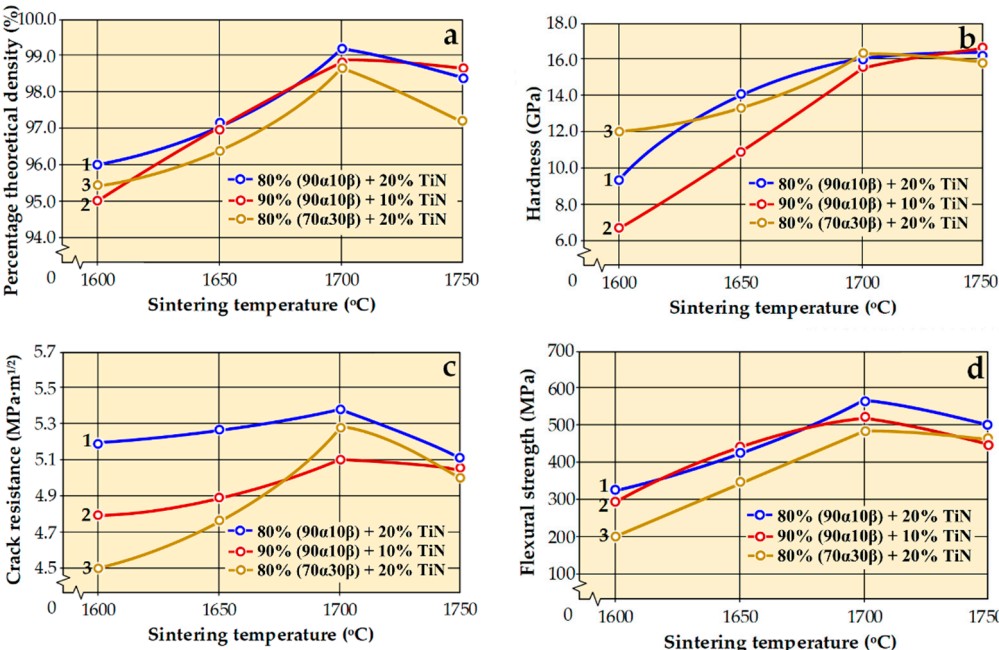

**Figure 9.** Dependences of the basic physical and mechanical properties of sintered ceramic samples made of various powder compositions such as 80% (90α10β) + 20% TiN (1), 90% (90α10β) + 10% TiN (2), and 80% (70α30β) + 20% TiN (3) from the spark plasma sintering temperature: (**a**) percentage theoretical density; (**b**) hardness; (**c**) crack resistance; and (**d**) flexural strength.

A general characteristic tendency of the temperature factor's influence at spark plasma sintering on the basic physical and mechanical properties is visible in Figure 9. The best values of the properties in all cases are achieved at a process temperature of 1700 °C. A further increase in temperature leads to a sharp decrease in the sintered material properties. The exception is hardness, which at 1750 °C remains at the same level for all specimens, Figure 9b. There is no doubt that a higher sintering temperature promotes recrystallization processes leading to rapid grain growth. The temperature of 1700 °C is a rational value for sintering all studied powder compositions based on SiAlON. Ceramics with a minimum number of pores are formed under these conditions judging by the fact that they have the maximum theoretical density (Figure 9a). It explains the maximum values at 1700 °C of crack resistance (Figure 9c) and flexural strength (Figure 9d).

Analysis of the basic physical and mechanical properties of the samples sintered from three versions of powder compositions shows that the best ratio of "theoretical density–hardness–crack resistance–flexural strength" in all cases had option (1): ceramic base α–β SiAlON 80 wt.% (α–SiAlON 90 wt.% + β–SiAlON 10 wt.%) + TiN 20 wt.% (composition code 80% (90α10β) + 20% TiN). The indicated samples at a temperature of 1700 °C have the following quantitative values of properties: theoretical density of 99.1%, hardness of 16.0 GPa, crack resistance of 5.3 MPa·m$^{\frac{1}{2}}$, and flexural strength of 550 MPa. It should be noted that the hardness of the experimental ceramics is in no way inferior to the industrially produced ceramics based on SiAlON, stabilized with $Yb_2O_3$ additives, but it is somewhat inferior to the best samples in terms of strength properties and crack resistance [30,37,39].

Summarizing the character of the experimentally obtained curves for various powder compositions (Figure 9), it can be concluded that the introduction of TiN particles in a volume of 10 wt.% during sintering (curve 2) does not provide an acceptable level of

properties of ceramics based on α–β SiAlON (mainly the above refers to crack resistance). In contrast, the introduction of TiN into the powder composition in a volume of 20 wt.% (curve 1) provides the best combination of properties. In this case, the SiAlON ceramic base's rational composition is the following ratio between the phases: α–SiAlON 90 wt.% and β–SiAlON 10 wt.%. It was shown experimentally that an increase in the β-SiAlON content to 30 wt.% in a ceramic powder composition noticeably decreases the sintered material's strength characteristics (curve 3).

Table 5 shows the results of an elemental quantitative EDX analysis of an experimental sample of 80% (90α10β) + 20% TiN and the ceramic material, from which commercial ceramic end mills are made, intended for the processing of nickel alloys, taken in the current study as a standard. The data of an experimental sample of 80% (90α10β) + 20% TiN showed the best combination of physical and mechanical properties in studies. As follows from the presented data, the experimental ceramics contains the following content of elements: 37.4% Si, 19.1% Ti, and 12.6% Al. Commercial ceramic significantly differs in elemental composition and contains: 45.7% Si, 7.0% Yb, and 6.5% Al. Furthermore, we studied two fundamentally different types of ceramics based on SiAlON.

**Table 5.** Results of elemental quantitative EDX analysis of sintered ceramics based on SiAlON.

| Specimen | Si | N | Ti | Al | O | Yb |
|---|---|---|---|---|---|---|
| Experimental 80% (90α10β) + 20% TiN | 37.4 | 19.9 | 19.1 | 12.6 | 11.0 | – |
| Commercial ceramics | 45.7 | 35.0 | – | 6.5 | 5.8 | 7.0 |

Figure 10 shows the results of microstructural analysis of fractures of two types of ceramics based on SiAlON, which demonstrate significant differences in the experimental sample 80% (90α10β) + 20% TiN (Figure 10a,c,e) and specimens of commercial ceramics (Figure 10b,d,f). It is natural since sintered ceramics' structure depends mainly on the type and number of sintering components and additives and the technological process parameters. Comparing scanning electron microscopy (SEM) images of the microstructures of the fracture surface of ceramic samples shows that both samples contain a significant amount of the α-modification of SiAlON as the dominant phase. This phase has high homogeneity and fineness for a specimen of commercial ceramics. Experimental ceramics are characterized by equiaxed and non-equiaxed grains with an average grain size of 3–5 μm, while commercial ceramics have a grain size of 1.5–3.5 μm. Besides, both samples have a certain content of the β-modification of SiAlON. This phase is more characteristic and noticeable in commercial ceramics since it has elongated (oblong) particles.

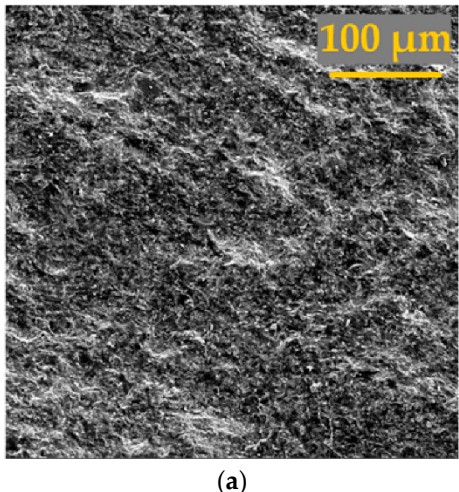
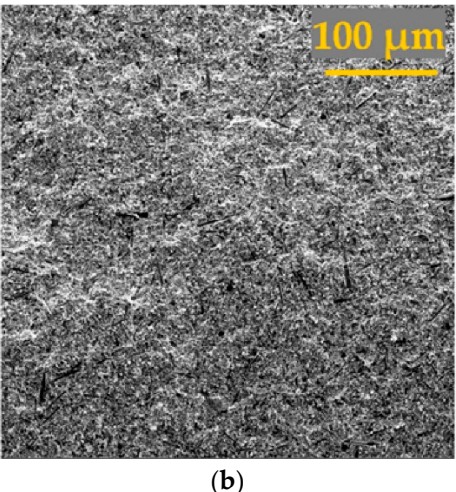

(**a**)  (**b**)

**Figure 10.** *Cont.*

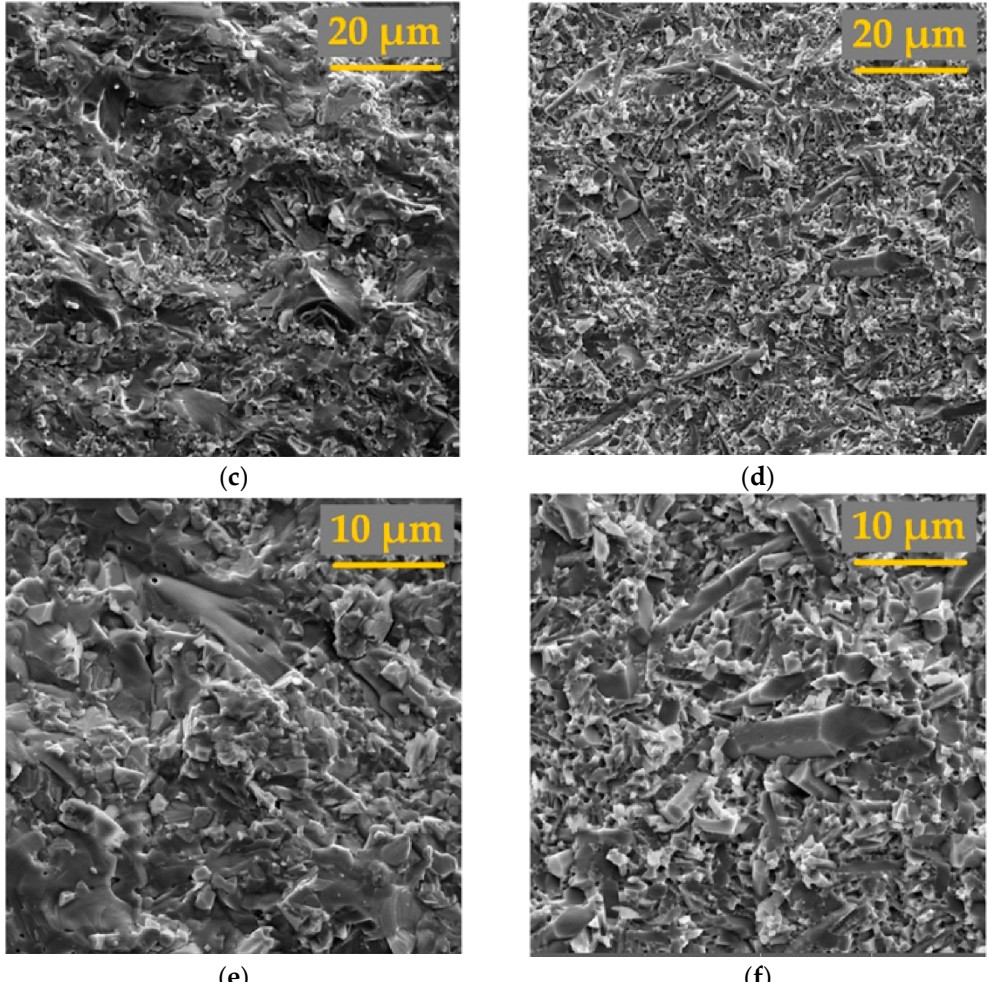

**Figure 10.** SEM images of fracture surface microstructures of sintered ceramic samples based on SiAlON: (**a**) experimental sample 80% (90α10β) + 20% TiN, ×1000; (**b**) specimen of commercial ceramics, ×1000; (**c**) experimental sample 80% (90α10β) + 20% TiN, ×5000; (**d**) specimen of commercial ceramics, ×5000; (**e**) experimental sample 80% (90α10β) + 20% TiN, ×10,000; and (**f**) specimen of commercial ceramics, ×10,000.

The elements distribution maps along the samples' fracture were obtained using EDX analysis for the possibility of a qualitative assessment of the uniformity of distribution in ceramic samples of the main elements and Ti- and Yb-containing phases (Figure 11). It can be seen that the Ti-containing phase is fairly uniformly distributed over the material volume after sintering in the experimental sample of 80% (90α10β) + 20% TiN (Figure 11a–c). Element distribution maps in a specimen of commercial ceramics (Figure 11d–f) demonstrate that the Yb-containing phase is uniformly distributed throughout the material structure. It can be concluded that the Yb-containing phase (stabilizing phase) is formed along the grain boundaries of ceramics based on SiAlON, and it is this that inhibits their growth during sintering and ensures the formation of a finer-grained and uniform structure in comparison with samples from experimental ceramics 80% (90α10β) + 20% TiN (Figure 10) based on the data of authoritative researchers [37–40].

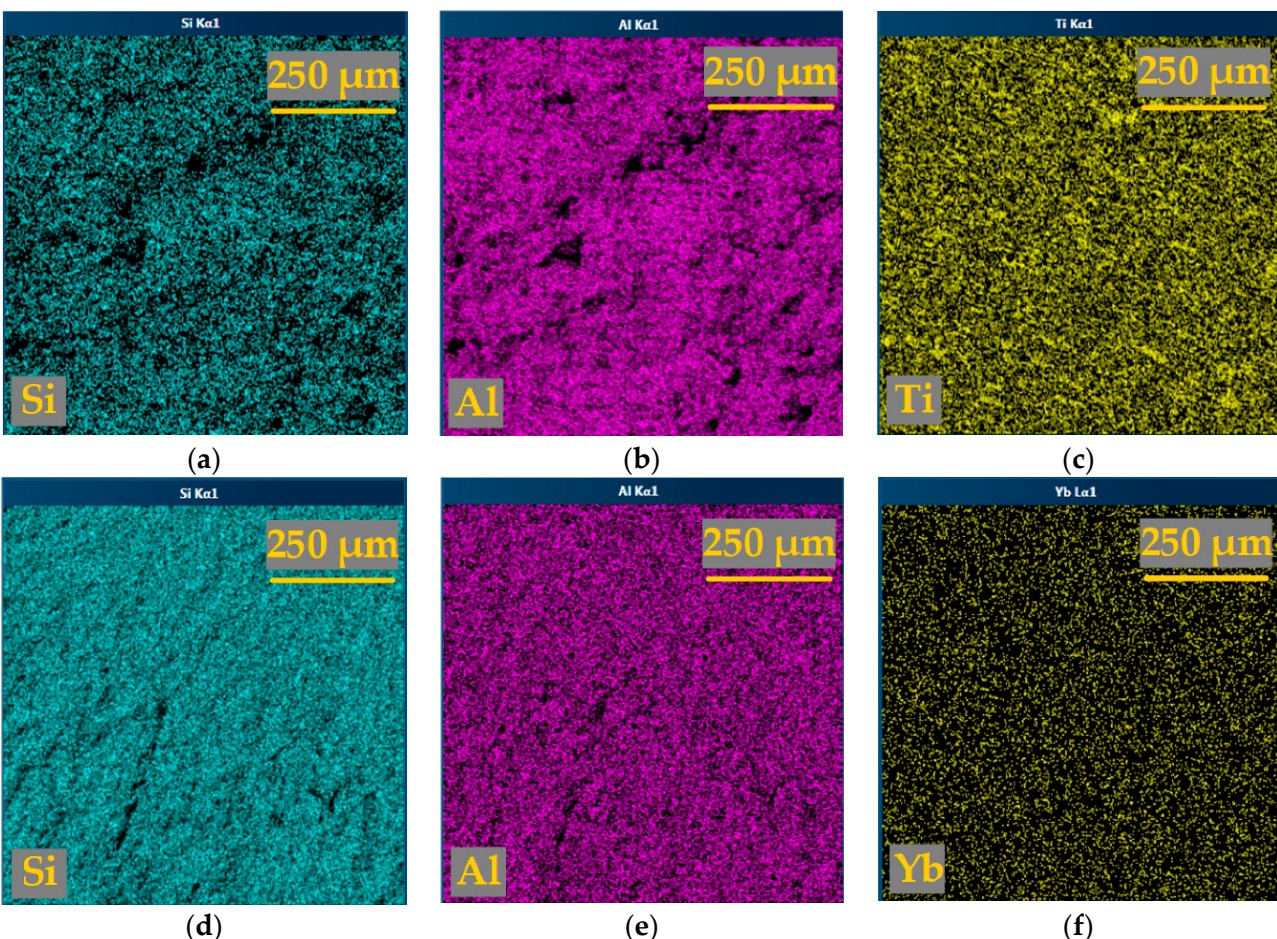

**Figure 11.** Distribution of chemical elements on the fracture surface of sintered ceramic material based on SiAlON samples, obtained by EDX analysis: (**a**) Si-content of experimental sample 80% (90α10β) + 20% TiN; (**b**) Al-content of experimental sample 80% (90α10β) + 20% TiN; (**c**) Ti-content of experimental sample 80% (90α10β) + 20% TiN; (**d**) Si-content of commercial ceramics specimen; (**e**) Al-content of commercial ceramics specimen; and (**f**) Yb-content of commercial ceramics specimen.

*3.2. Influence of DLC Coatings on the Transformation of the Sintered Ceramic Blanks Characteristics*

Figure 12 shows SEM images of the microstructure of thin sections of an experimental sample of 80% (90α10β) + 20% TiN ceramics (Figure 12a) and specimen of commercial ceramics (Figure 12b) after DLC coating deposition under identical technological conditions. Figure 12c presents an SEM image of a DLC coating's surface structure, and Figure 12d shows a TEM image of its volume structure. The coating thickness in total was 3.6 μm, including the thickness of 1.7 μm of the (CrAlSi)N sublayer and 1.9 μm of the functional DLC layer. It can be seen that the nitride sublayer has a columnar structure, while the outer DLC layer is characterized by an amorphous structure that microscopic methods cannot differentiate. The DLC layer does not have visible boundaries between grains in thickness, which are traditionally formed during the deposition of nitride coatings, while the surface of the DLC layer is a specific relief in the form of intergrown spherical segments of 0.2–1.5 μm in size. It should be noted that the formation of a characteristic relief is not affected by the ceramic base's surface layer's state before the coating deposition. The selected area diffraction (SAED) of the complex coating (Figure 12e,f) presents a complex phase of (CrAlSi)N sublayer and amorphous structure of DLC layers.

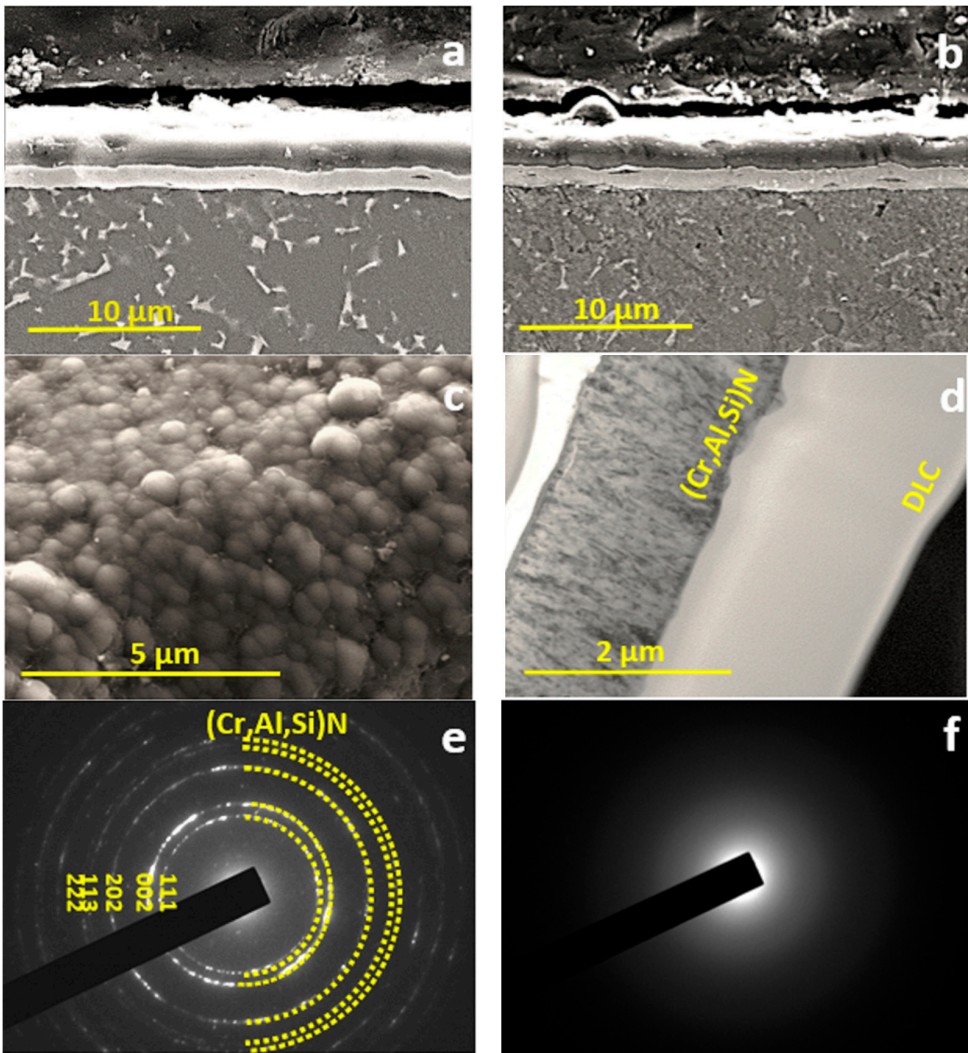

**Figure 12.** Microstructure of ceramic specimens with DLC coatings: (**a**) SEM image of a thin section of an experimental specimen 80% (90α10β) + 20% TiN ceramics with DLC coating; (**b**) SEM image of a thin section of commercial ceramics specimen with DLC coating; (**c**) SEM image of the DLC coating surface structure; (**d**) TEM image of the a two-layer DLC coating structure; (**e**) SAED of the (CrAlSi)N sublayer; and (**f**) SAED of the functional DLC layer.

Figure 13 shows the surface topographies of 80% (90α10β) + 20% TiN ceramic specimens before (Figure 13a) and after DLC coating deposition (Figure 13b). It is seen (Figure 13a) that the surface of the ceramic specimen includes a pronounced network of abrasive scratches, which are traditional for ceramic tools that are subjected to diamond sharpening. Besides, many craters from grains torn out during grinding are found on the ceramic surface previously shown in a series of previous works [17,21,25]. The noted features are defects of abrasive processing in essence, which are inherent in the process physics. The danger lies in the fact that ceramics are very sensitive to structural defects, which serve as stress concentrators and are often sources of tool's working surfaces spalling and chipping. It is possible to minimize or eliminate such defects in additional operations of finishing and polishing, which significantly increases the labor intensity and manufacturing cost of a ceramic product. The topography of the DLC-coated ceramic specimen surface (Figure 13b) demonstrates that the coating significantly modifies a thin surface layer's relief and significantly affects the size and shape of surface microroughness formed during the diamond sharpening stage. The coating fills the micro-grooves on the surface, thereby providing a kind of "healing" of the surface (it should be noted that a similar effect is observed when applying coatings of various compositions). There is reason to expect

that the coating will affect the characteristics of ceramic samples taking into account the described effect and the fact that the hardness of the formed DLC coating is not less than 25 GPa [73–77], which significantly exceeds the initial hardness of the tool ceramics, and the coating has a lower coefficient of friction and good strength of the adhesive bond with ceramic basis [78–80].

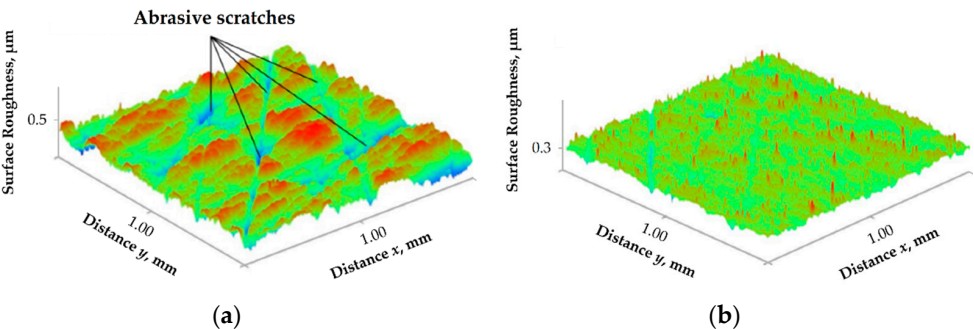

**Figure 13.** Topography of 80% (90α10β) + 20% TiN ceramic specimens surface areas: (**a**) before DLC coating deposition and (**b**) after DLC coating deposition.

Figure 14 shows histograms that illustrate the relationship between the applied critical load, at which destruction occurs, the path of punch movement along the z-axis until the destruction of the sintered ceramic samples, depending on the version of the powder composition. Data are given for samples before and after DLC coating deposition (quantification for each option was carried out based on four tests' results). The experimental data presented show that lower values of forces are required for the fracture of ceramic specimens 80% (70α30β) + 20% TiN and 90% (90α10β) + 10% TiN. It indirectly indicates their lower resistance to brittle fracture (this is confirmed by the studies presented in Figure 9). The coating has minimal effect on the increase in mean breaking force for these specimens. A different picture is observed for a sample of 80% (90α10β) + 20% TiN. Firstly, the average critical force is 5.2 kN, which is 10–25% higher than the corresponding indicator for the other two compositions. Secondly, the DLC coating deposition up to 6.8 kN (by 30%) increases the average critical force at which the sintered ceramic specimen breaks down (an increase in the punch travel path after which fracture occurs is also observed). The results obtained allow concluding that the coating can somewhat improve the fracture resistance of sintered ceramics based on SiAlON due to the transformation of the surface properties. However, it is only observed if the coating is applied to a ceramic base with a satisfactory combination of crack resistance and strength. It can be assumed that the improvements are a consequence of the effect mentioned above of "healing" of surface defects to a certain extent.

A series of tribological tests were carried out at a load of 1 N and a sliding speed of 10 cm/s according to the "ball-ceramic disk" scheme (due to the nickel alloy's increased wear counter body, an $Al_2O_3$ ball was used as a material in these tests) to assess the contribution of the coating to the change in the properties of the ceramics based on SiAlON (80% (90α10β) + 20% TiN) surface layer. Figure 15 systematized the tests' results at a friction length of 200 m before and after DLC coating deposition during tests without thermal exposure and under high-temperature heating conditions. The traditional character of the change in the friction coefficients of ceramic samples over time is observed in the room temperature results. At the very beginning of the tests, the uncoated ceramic sample has a friction coefficient of 0.2, which increases rather quickly and reaches 0.8 after 100 m of distance, which remains until the end of the tests. A very stable behavior of the coating during the entire test cycle, when the friction coefficient was invariably at the level of 0.1, is observed after DLC coating deposition. This coating belongs to the anti-friction class and is traditionally characterized by a reduced coefficient of friction. Considering that a ceramic cutter during the processing of a nickel alloy is subjected to high thermal effects, which has little in common with operating products at room temperature, the results of tribological

tests when heated to high temperatures are of the most significant interest. Under these conditions, SiAlON (80% (90α10β) + 20% TiN) samples without coating show unstable results, when the friction coefficient changes abruptly. Friction coefficient μ intensively increases after a value of 0.2 at the beginning and reaches a value of more than 0.7 after 25 m, then decreases to 0.45, and then it increases, decreases, and again increases, reaching more than 0.9 by the end of the tests. The authors described similar unstable results of other tool ceramics' behaviors (based on Al$_2$O$_3$) in a previously published study [25]. The DLC coating deposition onto ceramic samples strongly changes the conditions of frictional interaction: μ remains at a low level and varies slightly within the range of 0.09–0.15 for a sufficiently long time, and it begins to increase only after passing a distance of 150 m, reaching a value of 0.72 by the end of the tests. It can be assumed that the aforementioned is a consequence of the unique properties that a two-layer coating (CrAlSi)N/DLC possesses upon contact with the counter body under conditions of intense heat exposure. Superficial carbon DLC coating layers show poor results at increased thermal loads and often lose their initial microhardness, known from several authoritative works [12].

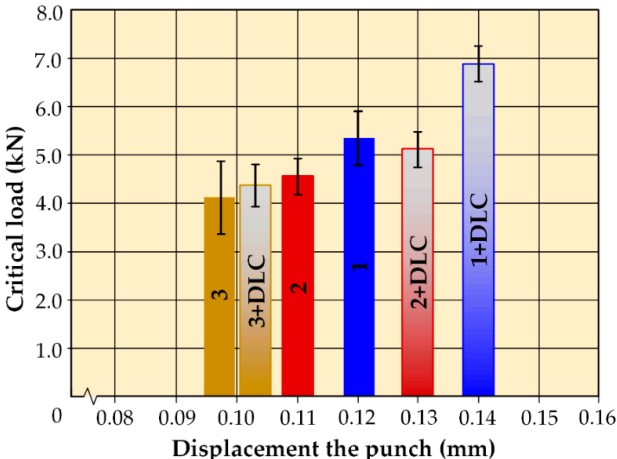

**Figure 14.** The relationship between the critical (breaking) load, the path of punch movement until the fracture of ceramic samples sintered from various powder compositions, where (1) is 80% (90α10β) + 20% TiN, (2) is for 90% (90α10β) + 10% TiN, and (3) is for 80% (70α30β) + 20% TiN before and after DLC coating deposition.

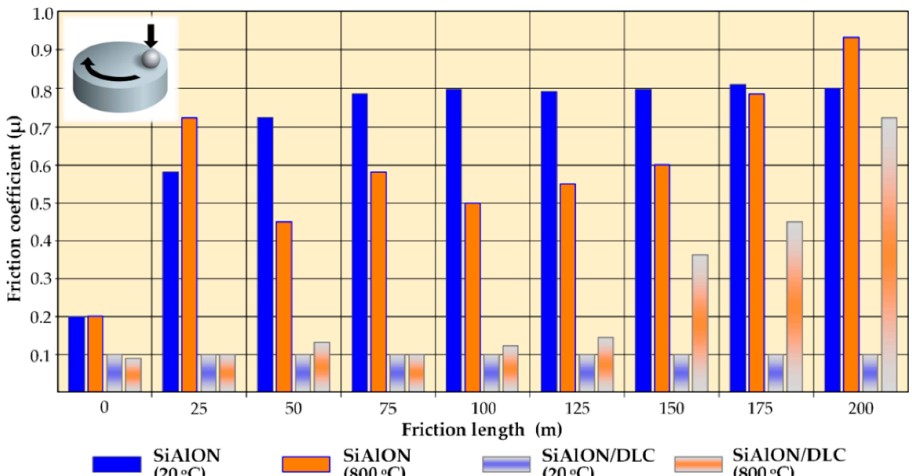

**Figure 15.** Dependences of the friction coefficient on the surface of experimental specimens made of SiAlON (80% (90α10β) + 20% TiN) ceramics on the friction path before and after DLC coating deposition under various test temperature conditions: without heating (20 °C) and with heating to 800 °C.

Nevertheless, due to the introduction of silicon into its composition during the deposition process, a formed DLC-Si layer makes it possible to significantly increase the DLC layer's thermal stability and expand the field of application of such coatings in this case [75,81,82]. In addition, the presence of a nitride sublayer under the DLC layer based on the Cr-Al-Si system increases the strength of the external DLC layer's adhesive bond with the tool base [83–85] and contributes to the formation of secondary wear-resistant phases during high-temperature heating [86–90]. When heated in oxygen, the coating components can form nonstoichiometric oxide phases, contributing to the change in the contact interaction between the ceramic article and the counter body nature [91–93].

### 3.3. Performance Testing of Solid Ceramic End Mills

The operational tests' main task was to compare the efficiency of two options of end mills: (1) experimental specimens of SiAlON (80% (90$\alpha$10$\beta$) + 20% TiN) ceramics obtained by spark plasma sintering and (2) industrial designs of commercial ceramic mills based on SiAlON, alloyed with stabilizing additives containing Yb. Another important task was to assess the prospects for using DLC coatings to increase these tools' wear resistance when processing heat-resistant nickel alloys. For the tests, groups of ceramic end mills of identical design were used (Figure 2). Section 2.7 details the performance test procedure, cutting data, and material properties. Figure 16 shows the dependences of flank wear on the operating time of two variants of ceramic end mills before and after DLC coating deposition. It can be seen that the experimental end mills developed within the framework of this work (1) are quite workable, but there is relatively intense wear of their cutting teeth, and they are noticeably inferior in operating time until critical wear (resistance) is reached compared to the reference specimens (2) commercial ceramic mills (5 min versus 7.1 min, respectively). Simultaneously, the spread of wear values for a ceramic end mill's teeth for commercial ceramic mills during operation is noticeably smaller. Such differences in the two types of ceramic end mills' tool life are not unexpected if we consider the experimental data presented in Figure 10. The comparative analysis of the microstructures of two variants of ceramics performed earlier showed that for the experimental samples sintered by the authors, there is a less uniform distribution of components over the volume of the sintered ceramic and a larger average grain size compared with commercial SiAlON ceramic doped with stabilizing additives containing Yb. It is well known that these characteristics are critical for tool ceramics and largely determine the tool's performance, especially under conditions of exposure to cyclic loads typical for milling. The DLC coating deposition makes it possible to reduce the wear rate of the experimental ceramic end mills (3), produced by the authors of the work, and demonstrates an increase in durability by 1.5 times, which is 7.7 min, relative to the uncoated cutting tool; at the same time, the spread of wear values on the ceramic end mill teeth slightly decreased. An important observation that can be made is that after DLC coating, the experimental end mills' durability is comparable to commercial ceramic mills' service lives. It cannot be considered an irregular or random result since the DLC coating deposition on commercial ceramic mills (4) demonstrates the same tendency: the coatings reduce the tool's wear rate and increase its tool life by 1.5, which is 11.2 min. Separately, it should be noted that the phenomena of pronounced chipping of cutting teeth of ceramic end mills were not observed in any of the tool's tested samples when carrying out resistance tests. The increase in the resistance of experimental samples of ceramic mills and commercial ceramic mills after DLC coating deposition established in this work is undoubtedly a consequence of the changes in the properties of the surface and surface layer of ceramics described in detail in Section 3.2.

Additionally, the work evaluated DLC coatings' effects on the surface quality (surface roughness) of machined nickel alloy NiCr20TiAl workpieces (Figure 17). The analysis of the presented curves, which were obtained based on the test results of ceramic end mills without coatings, shows that the height of the formed surface microroughness on the workpiece's surface with the tool operation course has a non-monotonic character

and characteristic peaks. Simultaneously, the curves obtained for experimental samples of ceramic end mills (1) and commercial ceramic mills (2) have a similar character. Considering that the level of antifriction properties of the tool surface strongly affects the quality of the workpiece surface layer, the nature of the experimental dependences of the processed workpiece surface roughness on the operating time is explained and in good agreement with the research results presented in Figure 15 (stepwise change with time in the friction coefficient of uncoated SiAlON-based ceramics under heating conditions up to 800 °C). Another regularity in the change in surface roughness can be observed during the operation of experimental (3) and commercial ceramic mills (4) after DLC coating deposition. The coating makes a pronounced contribution to improving the surface layer's quality of the workpiece being processed by changing the conditions of interaction with the cutting tool in the tribocontact zone. In the first minutes of operation, the surface roughness even slightly decreases and then begins to grow steadily. It is explained by the increase in the cutting tool wear zone and the actual area of its contact with the workpiece over time. Under such conditions, the intensity of their frictional and adhesive interaction increases, and the DLC coating contribution is leveled over time. Note that the described regularity correlates with the results of time variation of the friction coefficient SiAlON ceramic with a DLC coating under heating conditions up to 800 °C.

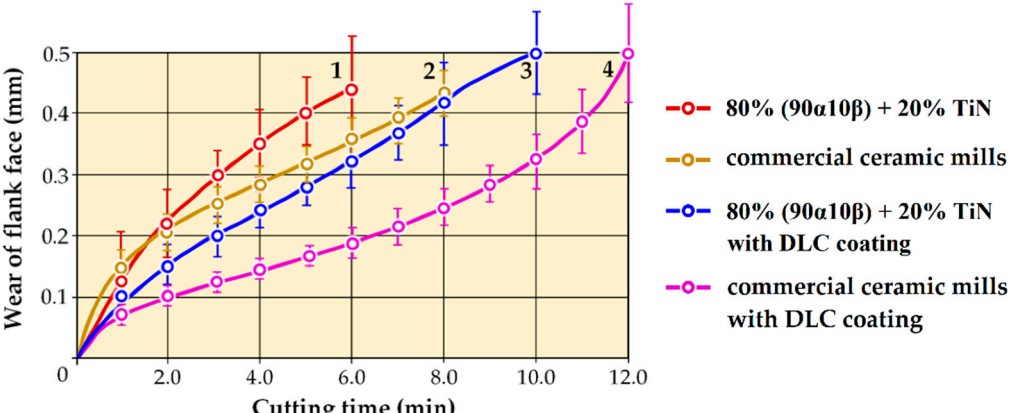

**Figure 16.** Dependence of flank face wear on the operating time of ceramic end mills before and after DLC coating deposition: (1) 80% (90α10β) + 20% TiN; (2) commercial ceramic mills; (3) 80% (90α10β) + 20% TiN with DLC coating; and (4) commercial ceramic mills with DLC coating.

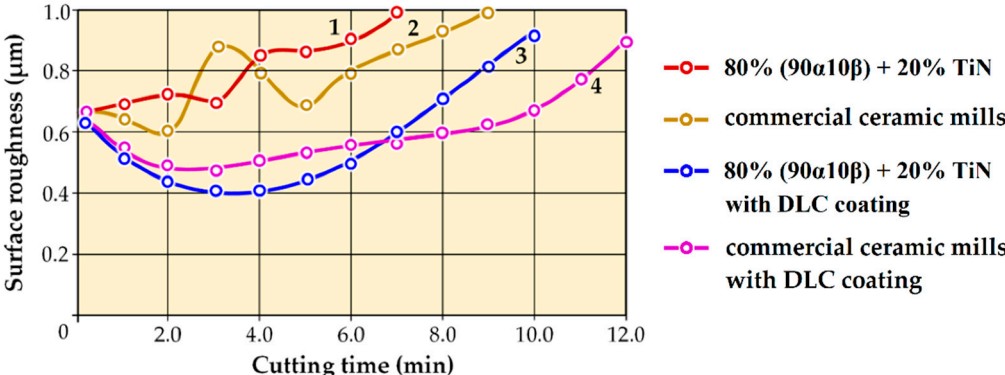

**Figure 17.** Dependence of the processed surface roughness on the operating time of ceramic end mills before and after DLC coating deposition: (1) 80% (90α10β) + 20% TiN; (2) commercial ceramic mills; (3) 80% (90α10β) + 20% TiN with DLC coating; and (4) commercial ceramic mills with DLC coating.

## 4. Conclusions

The complex of experimental studies carried out allows obtaining several original results that can be used to develop further the science-intensive direction associated with

the creation and maintenance of effective operating conditions for solid end mills made of ceramics based on SiAlON powder compositions for machining heat-resistant nickel alloys.

1. The rational quantitative ratio between the components of the $\alpha/\beta$-SiAlON-TiN powder composition for spark plasma sintering ceramics is $\alpha/\beta$-SiAlON 80 wt.% ($\alpha$-SiAlON 90 wt.% + $\beta$-SiAlON 10 wt.%) + TiN 20 wt.%. The temperature of 1700 °C is a rational value for the spark plasma sintering powder composition based on SiAlON since the samples made of experimental ceramics exhibit the best combination of theoretical density, crack resistance, and flexural strength at this value. The developed experimental ceramics are characterized by the presence in the structure of equiaxial and non-equiaxial grains with an average grain size of 3–5 μm compared to reference SiAlON-based commercial ceramics (including expensive Yb-containing stabilizing phases) that are used on an industrial scale with grain size of 1.5–3.5 μm. The grains of the commercial ceramics are more evenly distributed over the volume, and the structure is more homogeneous compared to developed ones. Everything indicates that the Yb-containing stabilizing phase is formed along the grain boundaries of the SiAlON-based ceramics (elemental EDX analysis of the fracture of the sample showed Yb of about 7.0%). This ceramic inhibits grain growth during sintering and ensures the formation of a finer-grained and uniform structure compared with samples from experimental $\alpha/\beta$-SiAlON-TiN ceramics. Nevertheless, the end mills of experimental ceramics are still efficient in processing the heat-resistant nickel alloy NiCr20TiAl at a cutting speed of 376.8 m/min. However, there is more intense wear of the cutting teeth, and they are noticeably inferior in durability to the reference samples of commercial ceramic end mills (5 and 7.1 min, respectively). Besides, experimental end mills have a wider spread of wear values on the tool's teeth during operation.

2. The deposition of a two-layer coating (CrAlSi)N/DLC with a thickness of about 3.6 μm significantly affects the properties of the surface and surface layer of sintered ceramics. The coating significantly modifies the relief of a thin surface layer and significantly affects the size and shape of surface microroughness and some defects formed at the stage of diamond sharpening. It somehow fills the micro-grooves on the surface, providing a kind of surface layer "healing." The experiments also show that the coating can somewhat improve the fracture resistance of sintered ceramics based on SiAlON due to the transformation of the surface properties, which manifests itself as an increase of up to 30% in the average critical force at which the fracture of the sintered ceramics occurs (5.2 kN for uncoated ceramics and 6.8 kN for ceramic coated (CrAlSi)N/DLC). However, the effect mentioned above is possible when the coating is applied to a ceramic base with a satisfactory combination of fracture toughness and strength.

3. Tribological tests have shown that (CrAlSi)N/DLC coating noticeably changes the conditions of frictional interaction of SiAlON-based ceramics with a counter body and provides a more stable ceramic sample surface layer operation under intense temperature exposure compared to ceramics without coatings. The friction coefficient for uncoated specimens changes abruptly and varies in the range from 0.2 to 0.9, showing instability throughout the entire friction distance. For the (CrAlSi)N/DLC coating, the friction coefficient remains at a low level for a relatively long time and varies slightly within the range of 0.09–0.15, and begins to increase only at the end of the tests, reaching 0.72.

4. Performance tests in laboratory conditions have shown that the application of (CrAlSi)N/DLC coating significantly affects the service life (durability) of solid ceramic end mills based on SiAlON powder compositions (both experimental end mill samples and commercial ceramic end mills) during processing heat-resistant nickel alloys such as NiCr20TiAl. The coating reduces the wear rate of experimental ceramic end mills relative to uncoated tools by a factor of 1.5 at a cutting speed of 376.8 m/min and provides durability of 7.7 min, which is comparable to the service life of commercial ceramic end mills. Furthermore, the application of a diamond-like carbon coating

on commercial ceramic mills demonstrates a similar result when the tool's wear rate decreases and its tool service life increases by 1.5 times and is 11.2 min. It is important to note that the coating contributes to improving the quality of the surface layer of the workpiece being processed by changing the conditions of interaction with the cutting tool in the tribocontact zone. In the first minutes of operation, the workpiece's surface roughness even slightly decreases and then begins to grow steadily. It is explained by the increase over time in the cutting tool wear area (land) and the actual area of its contact with the workpiece being processed. At the moment of failure, it is about 0.7 μm.

5.  Deposition of (CrAlSi)N/DLC coating can be considered a promising way to improve the performance of ceramic end mills. Nevertheless, it is impossible to cover all issues and study the behavior of solid end mills made of ceramics in a wide range of changes in operating conditions within the framework of one work. First of all, research is needed at increased cutting speeds, which should be the next step in developing this scientific direction since it is possible to test the efficiency of (CrAlSi)N/DLC coating under high-speed milling conditions of heat-resistant nickel alloys only experimentally. Besides, it is necessary to search and test the effectiveness of new options for coatings and other surface finishing methods of sintered ceramics, which could significantly improve the properties of the surface and surface layer of ceramic end mills and thereby positively affect the performance of the tool.

**Author Contributions:** Conceptualization, S.N.G.; methodology, M.A.V.; validation, S.V.F. and A.A.O.; formal analysis, P.M.P., P.Y.P. and A.E.; investigation, P.M.P., P.Y.P. and A.E.; resources, P.M.P. and P.Y.P.; data curation, A.A.O. and A.E.; writing—original draft preparation, M.A.V.; writing—review and editing, S.N.G.; visualization, A.A.O.; supervision, M.A.V.; project administration, S.N.G. All authors have read and agreed to the published version of the manuscript.

**Funding:** The study was supported by a grant of the Russian Science Foundation (project No. 21-79-30058).

**Institutional Review Board Statement:** Not applicable.

**Informed Consent Statement:** Not applicable.

**Data Availability Statement:** The data presented in this study are openly available in Figure 1a at (https://www.mmsonline.com/blog/post/applications-advance-for-solid-ceramic-end-mills, (accessed on 29 April 2021); https://www.cnctimes.com/editorial/solid-ceramic-endmills-from-kennametal-help-meet-critical-delivery-date, (accessed on 29 April 2021)), reference number [7,8]).

**Acknowledgments:** The work was carried out in the High-efficiency processing department of Moscow State University of Technology "STANKIN".

**Conflicts of Interest:** The authors declare no conflict of interest.

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
