# Peer review of "Development of DLC-Coated Solid SiAlON/TiN Ceramic End Mills for Nickel Alloy Machining: Problems and Prospects"

_coatings, doi:10.3390/coatings11050532_

Round 1
Reviewer 1 Report
In terms of scientific contributions, this is very interesting and thoroughly written research paper with a lot of investigation that has been done with quality methodology and advanced investigation techniques. The research results have great potential for practical application. The abstract accurately reflects the content. The research study methods are sound and accurate. The literature review and research study methods are explained clearly. The primary thesis, aims and objectives are discussed persuasively.
My only objection is related with the conclusions section, which are very long and wide, instead of being concise. My suggestion for authors is to shorten the conclusions and summarize the results and their contribution in a few short clues.
Author Response
Dear Reviewer,
Thank you very much for your kind evaluation of our work.
We would like to note that we appreciate your recommendation to shorten the conclusions and revised the manuscript according to your suggestion. The conclusions are shortened. We hope that, in the current version, the manuscript can be accepted for publication.
Kind regards,
Authors
Reviewer 2 Report
This study is Development of DLC-Coated Solid SiAlON/TiN Ceramic End Mills for Nickel Alloy Machining: Problems and Perspectives. This paper is worth to read and abstract is acceptable.
- It is well designed and will be interesting for the authors who use this technique.
- Experimental processes (experimental conditions) are organized.
- In spite of having some old references, it can be said that it is satisfactory.
- Overall, the space and arrangement between the figures (Tables) and the figure (Table) titles are organized. But the space arrangement between Table 4 and the paragraph below Table 4 don't do well.
- Two keywords (diamond-like coating, high speed cutting) do not appears in the article, and the other keywords (surface roughness) only appears in the description of Figure 18. Please modify.
- The conclusion has too many words and the discussion is too long, and it should be simplified.
- How to define “the wear of tooth’s flank face?” How to measure this value is equal to 0.4 mm and what is the size errors? Is the size errors high or low?
Author Response
Response to Reviewer 2 Comments
Dear reviewer,
Thank you very much for your kind evaluation of our work. We do agree with all your proposals and comments and have modified the manuscript according to them.
Introduced changes were marked by yellow in the text of the manuscript.
Kind regards,
Authors.
Point 1: It is well designed and will be interesting for the authors who use this technique.
Response 1: Thank you!
Point 2: Experimental processes (experimental conditions) are organized.
Response 2: Thank you!
Point 3: In spite of having some old references, it can be said that it is satisfactory.
Response 3: Thank you, we would like to note that more than 10% of the references are related to the works published in 2019-2021.
Point 4: Overall, the space and arrangement between the figures (Tables) and the figure (Table) titles are organized. But the space arrangement between Table 4 and the paragraph below Table 4 don't do well.
Response 4: Thank you very much for your kind suggestion. We have reorganized this fragment. We hope that it looks better in the current version of the manuscript.
Point 5: Two keywords (diamond-like coating, high speed cutting) do not appears in the article, and the other keywords (surface roughness) only appears in the description of Figure 18. Please modify.
Response 5: Thank you for noticing it. DLC is a commonly used acronym of diamond-like carbon coating. It appears in the article 69 times, including the title and three times in the abstract. It appears in the list of references eight more times. In modern coating physics and engineering, DLC-coating is the primary and prevailing name of diamond-like coatings in the titles of papers and as a part of the name of more complex coatings ((CrAlSi)N/DLC coating). https://en.wikipedia.org/wiki/Diamond-like_carbon
It is better to search with editor engine a collocation such as “diamond-like” or even “diamond like” to see all the primary meaning interpretations. We have modified a few places in the manuscript to appear often “diamond-like.”
High speed cutting appeared in various writing variations in the text as high-speed machining and high speed machining. Now it is unified: high-speed milling. It appears six times all over the text and shows graphs.
Surface roughness is modified as roughness to avoid this misunderstanding since it appears in the text 14 times in various ways: roughness parameter, microroughness. Now it is unified as surface roughness, surface microroughness, and surface roughness parameter.
Point 6: The conclusion has too many words and the discussion is too long, and it should be simplified.
Response 6: Thank you, the fragment is revised.
Point 7: How to define “the wear of tooth’s flank face?” How to measure this value is equal to 0.4 mm and what is the size errors? Is the size errors high or low?
Response 7: Thank you for the question on the merits of the experiment. The wear criterion of the tooth's flank face hf was measured according to the standard technique described in 2.7. Performance Testing of Solid Ceramic End Mills. Each end mill sample was researched optically every minute of machining (shown in Figure 17). When the wear area's critical size hf reaches 0.4 mm, the wear criterion of end mill samples was taken as tool failure. However, the experiments continued to hf reaches the range of 0.4 – 0.5 mm to demonstrate the dramatic tendency in tool wear. Measurement accuracy of ± 0.025 mm is 5.00 – 6.25 % of the measured value hf and less than standard accuracy tolerance in experiments. The fragment is revised.

Reviewer 3 Report
The article "Development of DLC-Coated Solid SiAlON/TiN Ceramic End Mills for Nickel Alloy Machining: Problems and Perspectives" by S. Grigoriev et al. is interesting and touches on the crucial subject of wear resistance of the machining tools. I found the article well planned and prepared. It provided good applicative knowledge and deserved to be published with minor corrections listed below:
- In part "2.3. Powder Compositions' Preparation and Sintering Ceramic Blanks" Authors should provide the temperature for the drying oven and details regarding the sieving process;
- In Table 3 Authors provide the results of the tribological tests – the information regarding the type of ceramic should be added;
- Please provide the details regarding TEM sample preparation methods;
- In part "3.1. Structure and Properties of Sintered Ceramic Blanks" I would expect to see results from X-ray diffraction experiments. The phase composition of the prepared ceramic should be examined;
- 9, Fig. 18 and Fig.18 – insert legend describing the curves for better clearness;
- 10 should be removed. It does not provide additional information – only the tables are interesting;
- In part "3.2. Influence of DLC Coatings on the Transformation of the Sintered Ceramic Blanks Characteristics" Authors claim that "the outer DLC layer is characterized by an amorphous structure" such claim should be supported by the SAED patterns. Please include electron diffraction patterns in Fig.13. Also, the presence of the (Cr,Al,Si)N phase should be supported by SEAD patterns.
- 136 – please insert description of the observed layers in a and b part for better clearness;
- Fig 14 – the z scale (colour coding) should be provided;
Author Response
Response to Reviewer 3 Comments
Dear reviewer,
Thank you very much for your kind evaluation of our work. We do agree with all your proposals and comments and have modified the manuscript according to them.
Introduced changes were marked by green in the text of the manuscript.
Kind regards,
Authors.
Point 1: In part "2.3. Powder Compositions' Preparation and Sintering Ceramic Blanks" Authors should provide the temperature for the drying oven and details regarding the sieving process;
Response 1: Thank you for your kind comment and suggestions. We have added relevant data to the text of the manuscript.
Point 2: In Table 3 Authors provide the results of the tribological tests – the information regarding the type of ceramic should be added;
Response 2: Thank you, the relevant data is added.
Point 3: Please provide the details regarding TEM sample preparation methods;
Response 3: Thank you, the relevant data is added to the paragraph.
Point 4: In part "3.1. Structure and Properties of Sintered Ceramic Blanks" I would expect to see results from X-ray diffraction experiments. The phase composition of the prepared ceramic should be examined;
Response 4: Thank you for your kind suggestion. We apologize since it was beyond the scope of the study (Development of DLC-Coated Solid SiAlON/TiN Ceramic End Mills), and we did not want to overload the article since it is already 28 pages. We aimed to present a new way of DLC-Coated Solid SiAlON/TiN Ceramic End Mill production and not develop only ceramics for tool applications. However, we can conduct it further as a part of another article. The article of our research group related to more comprehensive research of material is following https://www.mdpi.com/2076-3417/11/2/657
Point 5: 9, Fig. 18 and Fig.18 – insert legend describing the curves for better clearness;
Response 5: Thank you for your kind proposal. We have added the legend to Figures 9, 17, 18.
Point 6: 10 should be removed. It does not provide additional information – only the tables are interesting;
Response 6: Thank you, the figure is revised.
Point 7: In part "3.2. Influence of DLC Coatings on the Transformation of the Sintered Ceramic Blanks Characteristics" Authors claim that "the outer DLC layer is characterized by an amorphous structure" such claim should be supported by the SAED patterns. Please include electron diffraction patterns in Fig.13. Also, the presence of the (Cr,Al,Si)N phase should be supported by SEAD patterns.
Response 7: Thank you, we have added the relevant images of SAED patterns (Figure 12 e,f) that confirm the amorphous structure of the coating functional layer.
Point 8: 136 – please insert description of the observed layers in a and b part for better clearness;
Response 8: Thank you, the figure is revised.
Point 9: Fig 14 – the z scale (colour coding) should be provided;
Response 9: Thank you, the figure is revised. We should add that in the context of the article, it has no principal character. The 3D profiles were aimed to show surface morphology – the presence of scratches and the healing effect of the coating.
